# Deep Learning for Osteoporosis Classification Using Hip Radiographs and Patient Clinical Covariates

**DOI:** 10.3390/biom10111534

**Published:** 2020-11-10

**Authors:** Norio Yamamoto, Shintaro Sukegawa, Akira Kitamura, Ryosuke Goto, Tomoyuki Noda, Keisuke Nakano, Kiyofumi Takabatake, Hotaka Kawai, Hitoshi Nagatsuka, Keisuke Kawasaki, Yoshihiko Furuki, Toshifumi Ozaki

**Affiliations:** 1Department of Orthopaedic Surgery, Kagawa Prefectural Central Hospital, Takamatsu, Kagawa 760-8557, Japan; lovescaffe@yahoo.co.jp (N.Y.); kkeisuke@a1.mbn.or.jp (K.K.); 2Department of Oral and Maxillofacial Surgery, Kagawa Prefectural Central Hospital, 1-2-1, Asahi-machi, Takamatsu, Kagawa 760-8557, Japan; furukiy@ma.pikara.ne.jp; 3Department of Oral Pathology and Medicine, Graduate School of Medicine, Dentistry and Pharmaceutical Sciences, Okayama University, Okayama 700-8525, Japan; keisuke1@okayama-u.ac.jp (K.N.); gmd422094@s.okayama-u.ac.jp (K.T.); de18018@s.okayama-u.ac.jp (H.K.); jin@okayama-u.ac.jp (H.N.); 4Search Space Inc., Tokyo 151-0072, Japan; akira@searchspace.cloud (A.K.); ryosuke@searchspace.cloud (R.G.); 5Department of Musculoskeletal Traumatology, Okayama University Graduate School of Medicine, Dentistry and Pharmaceutical Sciences, Okayama 700-8558, Japan; tnoda@md.okayama-u.ac.jp; 6Department of Orthopaedic Surgery, Okayama University Graduate School of Medicine, Dentistry and Pharmaceutical Sciences, Okayama 700-8558, Japan; tozaki@md.okayama-u.ac.jp

**Keywords:** osteoporosis, deep learning, hip radiograph, ensemble model

## Abstract

This study considers the use of deep learning to diagnose osteoporosis from hip radiographs, and whether adding clinical data improves diagnostic performance over the image mode alone. For objective labeling, we collected a dataset containing 1131 images from patients who underwent both skeletal bone mineral density measurement and hip radiography at a single general hospital between 2014 and 2019. Osteoporosis was assessed from the hip radiographs using five convolutional neural network (CNN) models. We also investigated ensemble models with clinical covariates added to each CNN. The accuracy, precision, recall, specificity, negative predictive value (npv), F1 score, and area under the curve (AUC) score were calculated for each network. In the evaluation of the five CNN models using only hip radiographs, GoogleNet and EfficientNet b3 exhibited the best accuracy, precision, and specificity. Among the five ensemble models, EfficientNet b3 exhibited the best accuracy, recall, npv, F1 score, and AUC score when patient variables were included. The CNN models diagnosed osteoporosis from hip radiographs with high accuracy, and their performance improved further with the addition of clinical covariates from patient records.

## 1. Introduction

Approximately 50% of women and 20% of men will experience osteoporotic fractures in their lifetime [1]. The standard test for diagnosing osteoporosis is the estimation of bone mineral density (BMD) in the proximal femur and lumbar spine with dual-energy X-ray absorptiometry (DXA). The US Preventive Services Task Force has recommended screening for osteoporosis with BMD to prevent osteoporotic fractures in women 65 years and older [2]. The early diagnosis of osteoporosis is important for the prevention of osteoporotic fractures because therapeutic drug treatments fractures, because treatments are more effective in the early stages of the condition, before fractures have appeared [3]. One disadvantage of using DXA is the occurrence of relevant measurement errors that caused by the surrounding soft tissues [4,5]. Other disadvantages include radiation exposure and high device costs [6,7]. Easy access to bone densitometry examinations is important, particularly in developing countries.

Osteoporosis is an insidious skeletal disease in which the density (and calcium content) of the bones decreases; X-rays can aid in its detection [8]. The Singh index is a typical classification system for the bone density of the femoral neck based on the qualitative visibility of trabecular types [9]. On the other hand, quantitative assessment using hip X-ray imagery, such as estimating the ratio of the cortical width to the total width, has been performed since approximately 1970 [10]. The cortical-thickness index has been found to be significantly correlated with the BMD at the femoral neck and with predictions of the fracture-risk assessment tool [11,12]. Moreover, the canal bone ratio exhibits a strong correlation with the T-score and has the best overall performance in diagnosing osteoporosis with receiver operating characteristic (ROC) curve analysis [13]. In this manner, clinicians have diagnosed osteoporosis using the bone morphology of hip radiographs.

Artificial intelligence (AI) has been used as an adjunct technology for imaging interpretation, and as an initial screening tool [14]. A 2019 review article reported that recent developments in AI have led to successful applications aiding osteoporosis diagnosis [15]. The following modalities have been used: dental radiographs [16,17], spine radiographs [14,18], hand and wrist radiographs [19,20,21,22], DXA imaging [23,24], and spine computed tomography [25,26]. However, two reports are available on osteoporosis diagnosis from hip radiographs using machine learning [27,28]. To our knowledge, no study has yet reported on osteoporosis diagnosis based on hip radiographs using deep learning (DL).

Image-only models can be used to augment human interpretations. Although DL studies on various image-based osteoporosis-prediction methods have been published, these did not consider patient covariates, which clinicians use when estimating the pre-test probability during the decision process. Patient and healthcare process variables impress patterns into observational healthcare data, and these patterns can be learned by statistical learning algorithms [29]. Hip fracture diagnosis was predicted moderately well from images alone (area under the curve (AUC) = 0.78), and the prediction was improved by combining image features with patient data (AUC = 0.86) [29]. The direct extension of DL image models with known covariates can improve the model performance, which would be extremely useful in a clinical setting.

We hypothesized that combining image features with patient data would improve the accuracy of osteoporosis diagnosis. The purpose of this study was to diagnose osteoporosis with hip radiographs using DL, and to investigate whether adding clinical data would improve the diagnostic performance compared to the image mode alone.

## 2. Materials and Methods

### 2.1. Study Design

The purpose of this study is to classify the presence or absence of osteoporosis from hip radiographs of the hip joint. We used a segmented dataset to mimic the diagnostic range of osteoporosis in the DXA method. In addition, clinical covariates extracted from clinical records were added to the dataset. Several different convolutional neural networks (CNNs) were used to diagnose osteoporosis from hip radiographs. We investigated the accuracy of the predictive diagnosis of osteoporosis from the combination of clinical covariates with the various CNN models.

### 2.2. Data Acquisition

We retrospectively used clinical and image data from March 2014 to December 2019. This study was approved by the Institutional Review Board of the Kagawa Prefectural Central Hospital (Approval No. 894). This study involved 1223 consecutive patients 60 years of age or older who had hip radiographs and who underwent DXA at our hospital six months before and after the date that the hip radiograph was performed.

In this study, the term “hip fracture” refers to a femoral neck or trochanteric fracture.

We excluded the following images: 50 images of osteoarthritis with femoral head deformity, 32 unclear or poor images, 7 images showing artificial objects made of materials such as metal, 2 showing femoral bone deformity following prior fractures, and 1 showing pathological fracture. Thus, 1131 hips (708 fractured and 423 non-fractured) remained for further analysis.

### 2.3. Data Preprocessing

Simple hip radiographs of each patient were used to acquire the images. All images were output in TIFF format (size: 2836 × 2373 pixels) from the Kagawa Prefectural Central Hospital PACS system (HOPE Dr ABLE-GX, FUJITSU Co., Tokyo, Japan). For the images, we performed segmentation of the hip joint area. Six orthopedic surgeons manually placed and cropped areas of interest on the X-ray images using Photoshop Elements (Adobe Systems, Inc., San Jose, CA, USA) under the supervision of an orthopedic specialist (N.Y., an author of this paper).

The side of the hip measured using DXA was selected as the cropped side in the pre-analysis image-cropping method. The lines of the femoral head and the inferior margin of the lesser trochanter were selected and cropped to include the lines of the femoral head and the inferior margin of the lesser trochanter, as illustrated in Figure 1, as the range cropped with the DXA measurement. The cropped area imitated the osteoporosis diagnosis range obtained using the DXA method. The cropped images were saved in PNG format. None of the orthopedic surgeons who performed the cropping were informed of the patient’s BMD status.

### 2.4. Diagnosis of Osteoporosis

BMD measurements (g/cm^2^) were performed at the hip through DXA (HOLOGIC Horizon-A, Apex software version 13.6.0.4, Bedford, MA, USA) by trained personnel using equal measurement routines. The standard position was used and the scanned image adhered to the following criteria [30]: the hip joint was located in the center of the image, with 15° to 25° internal rotation, with the femur neck, head, and greater trochanter completely in the image. The measurement was normally performed at the left hip. When the left hip had a metal implant or a high degree of deformity, the right hip was selected. The investigated parameters included the BMD (g/cm^2^) and T-score, which were generated automatically. Osteoporosis was diagnosed if the T-score of the bone density obtained by DXA was less than −2.5, following World Health Organization diagnostic criteria [31].

### 2.5. Clinical Covariates

Osteoporotic fractures are common fracture types, and age, gender, and body mass index (BMI) are clinically important risk factors [32]. Four clinical covariates types, namely age, sex, BMI, and history of hip fracture, were accordingly selected from the clinical factors of patients with osteoporosis in the study. The BMI is the weight in kilograms divided by the square of the height in meters (kg/m^2^). The height and weight were recorded at the time of the BMD measurement. A history of hip fracture was also selected as a patient factor because hip fractures affect the secondary fracture among osteoporotic fractures [33].

The clinical and demographic characteristics of patients in the hip radiographic image dataset in this study are presented in Table 1.

### 2.6. CNN Model Architecture

The performance of the following CNN models was evaluated: (1) ResNet18 [34], (2) ResNet34 [34], (3) GoogleNet [35], (4) EfficientNet b3 [36], and (5) EfficientNet b4 [36]. We selected ResNet and GoogleNet, the former winners of the ImageNet Large Scale Visual Recognition Challenge, a competition of computer-based image recognition models, as the CNNs. We also used EfficientNet, a state-of-the-art CNN model that is relatively smaller and faster than the other CNN models.

All CNNs were used for transfer learning with fine-tuning because a pretrained model reduces the training time and the number of images required to create a suitable classifier [37]. The DL process was performed with the Python package PyTorch.

After creating the branches of the image data with the CNN, a two-layered perceptron was created for the clinical covariates branches. Finally, the image data processed by the CNN were combined with the clinical covariates and output as a fully connected layer.

The dataset was randomly divided into training, validation, and testing sets in a ratio of 8:1:1. This study was performed using a Tesla P100 graphical processing unit. Various data augmentation techniques were used to prevent overfitting owing to the small dataset size. The training images were randomly rotated from −25 to 25, with a 50% chance to flip vertically and 50% chance to flip horizontally. Moreover, the darkness was randomly changed by −5% to 5% and the contrast by −5% to 5%. During learning, data augmentation was applied only to the training image data when the images were taken out in batches. Each training image was processed with a 50% chance of data augmentation. The optimizer, weight decay, and momentum were common to all the CNNs. In this study, the optimizer used stochastic gradient descent, and the weight decay and momentum were 0 and 0.9, respectively. A learning rate of 0.01 was used for EfficienctNet b3 and b4; 0.001 was used for the other CNNs. All the models analyzed a maximum of 100 epochs. We used the early stop method to terminate data augmentation if the validation error did not update 20 times in a row.

### 2.7. Architecture of the Ensemble Model

The ensemble model combines the above CNN mentioned model with structural data and tries to classify the diagnosis of osteoporosis. As a preparation, we preprocessed the structural data. Age and BMI translate mean normalization, and sex and history of fractures translated into a One-hot vector representation. As a result, 1 × 6 dimensional vectors were created. Extracted from the convolutional layers in the CNN of the image, the one-dimensional reshaped result and the 1 × 6 dimensional data created from the structural data were combined.

Finally, these results were put into the fully connected layer, and the prediction of the final training model was output (Figure 2).

### 2.8. Performance Metrics

The CNN models used in the modality-specific transfer learning with fine-tuning and ensemble learning were evaluated in terms of the following performance metrics: (1) accuracy, (2) precision (positive predictive value; ppv), (3) recall (sensitivity), (4) specificity, (5) negative predictive value (npv), (6) F1 score, and (7) AUC score.
(1)$$\mathrm{accuracy}=\frac{\mathrm{TP}\;+\;\mathrm{TN}}{\mathrm{TP}\;+\;\mathrm{FP}\;+\;\mathrm{TN}\;+\;\mathrm{FN}}$$
(2)$$\mathrm{precision}=\frac{\mathrm{TP}}{\mathrm{TP}\;+\;\mathrm{FP}}$$
(3)$$\mathrm{recall}=\frac{\mathrm{TP}}{\mathrm{TP}\;+\;\mathrm{FN}}$$
(4)$$\mathrm{specificity}=\frac{\mathrm{TN}}{\mathrm{TN}\;+\;\mathrm{FP}}$$
(5)$$\mathrm{negative}\;\mathrm{predictive}\;\mathrm{value}=\frac{\mathrm{TN}}{\mathrm{TN}\;+\;\mathrm{FN}}$$
(6)$$F1\;\mathrm{score}=2\times \frac{\mathrm{precision}\;\times \;\mathrm{recall}}{\mathrm{precision}\;+\;\mathrm{recall}}$$

TP and TN indicate the number of true positive and true negative results, respectively, whereas FP and FN indicate the number of false positives and negatives. We also calculated the ROC curve and measured the AUC.

### 2.9. Visualization of Computer-Assisted Diagnostic System

Guided Grad-CAM provides a direct visualization of the values in a map, and is a combination of the Grad-CAM [38] and back-propagation visualization [39] techniques. It shows information that is significant for classification: the high gradient of the input to the last convolutional layer. In this study, the heat map visualizations were displayed relative to the range of values in the image. All the visualizations were performed using iridescent map projections. Within the region of interest, high attenuation was shown in green and low attenuation in red.

## 3. Results

### 3.1. Prediction Performance

#### 3.1.1. Hip Radiographic Image

The performance evaluation of the five CNN models using only hip radiographs is presented in Table 2. EfficientNet b3 and GoogleNet were unsurpassed in accuracy (0.8407), precision (0.8929), and specificity (0.8824). ResNet34 was unsurpassed in accuracy (0.8407), F1 score (0.8500), and AUC score (0.9203). EfficientNet b4 was the best CNN model in recall (0.8548) and negative predictive value (0.8085).

#### 3.1.2. Hip Radiographic Image-Connected Clinical Covariates

Table 3 presents the performance evaluation for hip radiographs combined with clinical covariates analysis. EfficientNet b3 was unsurpassed in accuracy (0.8850), recall (0.8871), negative predictive value (0.8654), F1 score (0.8943), and AUC score (0.9374). ResNet34 exhibited the best precision (0.9273) and specificity (0.9216). Very interestingly, the addition of clinical covariates resulted in an increase in almost all performance metrics in all CNN networks over the analysis of hip radiographs alone (Table 4). Figure 3 shows the ROC curves for both cases.

### 3.2. Visualization of Model Classification

Figure 4 depicts the focused visualization area obtained by Guided Grad-CAM. We selected the ensemble analysis (clinical covariates added to the hip radiographs data) using EfficientNet b3 and EfficientNet b4 with the good performance metrics. In the radiographs of both osteoporosis and normal patients, the relatively distinct areas of shading around the lesser trochanter were identified as deep-learned feature areas.

## 4. Discussion

This study demonstrated that CNNs can diagnose osteoporosis from hip radiographs with relatively high accuracy. Moreover, including patient variables involved in routine clinical setting improved the accuracy of predictions compared to those using the image model alone. The EfficientNet b3 network exhibited the highest performance (accuracy: 88.5%; recall: 88.7%; F1 score: 0.8943; and AUC score: 0.9374) in diagnosing osteoporosis from proximal femoral cropped images from entire hip radiographs among the five networks.

The advantages of our study over previous studies that diagnosed osteoporosis using DL [27] are that it was based on a larger number of cases, and included more clinically suspicious cases and patients considered at risk of developing osteoporosis in the future for osteoporosis. The addition of patient variables offered important information, which improved the sensitivity and AUC score in particular. Our classification model combines image features and patient factors with the help of a neural network. We conclude that the diagnostic accuracy improved because we could make inferences while simultaneously considering vital information related to clinical covariates that cannot be extracted from images alone. To the best of our knowledge, this was the first study to compare the relative osteoporosis diagnostic efficiencies of DL systems that use images alone and those that also use patient variables. The results with relative high sensitivity suggest that the model of the image with patient variables offers a superior tool for screening osteoporosis in a clinical setting.

The EfficientNet b3 network was the highest accuracy of all the CNNs used in this study. The diagnostic accuracy varied significantly depending on the difference among the CNNs. We speculated that factors such as the image quality and quantity would directly affect the analytical findings. The worst classification performance was that of the several basic CNNs with relatively few convolutional layers. It can be inferred that CNN models with few convolutional layers have limited machine-learning capability for binary classification (in our case, osteoporosis or not) [40].

In this study, the performance of the CNNs was comparable to the diagnostic accuracy in a similar previous studies using hip radiographs with AUC 0.50–0.86 [27,28]. The area identified by Grad-CAM also has some test images that are somewhat consistent with the area around the lesser trochanter, which may allow the diagnosis of osteoporosis. The output of the Grad-CAM is purely for reference and does not directly imply the prediction itself [17]. Because our model has been trained for classification but not for localization, it might focus on outside femoral bone as shown by the Grad-CAM. Nonetheless, it is important to note that the prediction is correct. In the future, it will be necessary to study a higher number of cases and examine images of various sizes as well as images of soft-tissue contrast.

Although plain hip radiographs are the preferred approach for diagnosing hip fractures, they not the gold-standard imaging modality for osteoporosis detection. However, the accuracy for osteoporosis diagnosis of hip radiographs with DL in this study was comparable to that of fracture detection reports based on hip radiographs with DL [41,42,43]. These studies reported that the range of precision is 90.6–95.5%. Therefore, when hip fractures are diagnosed with hip radiographs, it is possible to diagnose osteoporosis simultaneously. The strategy of fix and treat, with surgical fixing of the fracture site and drug treatment for osteoporosis, should become routine in trauma management [44]. With the DL technique, this can be realized using only hip radiographs and a few patient variables.

The proposed computer-assisted diagnostic system is advantageous owing to its low cost, widely available radiographic modality, and simple protocol. Furthermore, it reduces measurement errors in the conventional objective assessment of Singh’s index [9], and those resulting from mal-rotation of the hip when interpreting BMD values [45]. A previous study on the effect of leg rotation emphasized the need for proper positioning of the hip during DXA scanning [46]. This is one of the unsolvable problems of DXA that may be avoided using the DL technique.

In our study, using structured patient covariable data with the images was more efficient for diagnosing osteoporosis than using the images alone. The patient clinical covariates were found to be the key features for osteoporosis diagnosis. The results were consistent with those of previous studies [32,47]. A few studies have been conducted on osteoporosis diagnosis with image and clinical covariates using DL. An artificial neural network model using clinical symptoms and image features extracted from lumbar radiographs exhibited an accuracy of 95.8% and an AUC score of 0.95 [48]. The review reported that the range of precision is 76.7–97.9% [49]. In our study, DL using hip radiographs and patient variables yielded results that were comparable in diagnostic accuracy to osteoporosis diagnosis by DL using only patient variables [49]. We speculated that adding clinical covariates to the image models would improve their performance in terms of the information amount and quality. More complex CNN models may assign greater importance to clinical covariates than to image information.

Several recommendations can be made for further research. In this study, the images were cropped to include the femoral head as a preprocessor to diagnose osteoporosis. In the future, it would be desirable to create a network that can automatically detect osteoporosis by detecting the site of the hip joint from uncropped hip radiographs. The next challenge will be to apply the technique to diagnosing osteoporosis by simultaneously detecting the target hip joint from hip radiographs using various networks, such as region-based CNNs, you only look once, and the single shot multibox detector.

This study has certain limitations. Owing to the narrow selection of CNN models, there may be differences in the availability of clinical covariates as a result of variations in the CNN structure. New DL algorithms with deep and wide layers or modified stratification methods are being developed continually [50]. We need to select an appropriate model for high-quality images that can also handle clinical covariates. It will thus be necessary to study various other CNN models in the future. Furthermore, it was difficult to collect a sufficient amount of image data from a single general hospital. CNNs with less data can lead to overfitting. In general, deep CNN models trained from pretrained deep neural networks on large image datasets are effective for general image classification. The accuracy of CNN classification diagnosis would be improved by increasing the number of images in a multicenter study. Finally, only four patient variables were considered in this study. Many confounding factors exist for osteoporosis diagnosis in patients [49]. In a previous comprehensive study, the somatic factors selected by the support vector machine were age, height, weight, BMI, duration of menopause, duration of breastfeeding, estrogen therapy, hyperlipidemia, hypertension, osteoarthritis, and diabetes mellitus [51]. Also, the bone remodeling rate can be assessed by the measurement of bone turnover markers [52]. Blood sampling is thus a candidate patient variable. Furthermore, BMD and hip geometry vary according to race and gender. For example, compared to other races, Asians have thicker cortical bones and lower buckling ratios [53,54]. Big data and additional patient variables will enable greater accuracy and aid further research.

## 5. Conclusions

We demonstrated that particular deep CNNs could diagnose osteoporosis from hip radiographs with high accuracy. In particular, the pretrained EfficientNet b3 exhibited the most accurate performance among the CNNs we considered. Moreover, adding patient variables as clinical covariates improved the sensitivity and AUC score for osteoporosis diagnosis. These results may play an important role in osteoporosis diagnosis in a clinical setting.

## Figures and Tables

**Figure 1 biomolecules-10-01534-f001:**
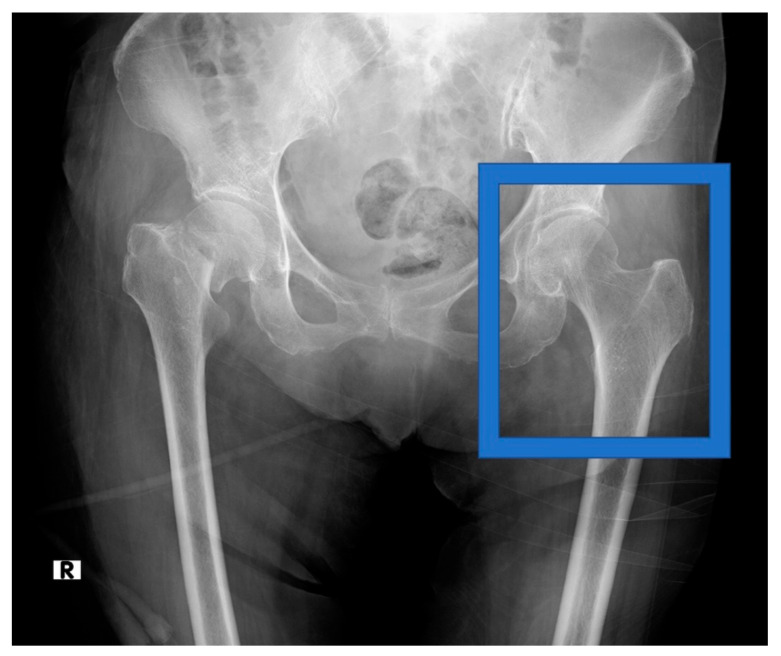
Hip radiograph before analysis, showing region of interest that was cropped.

**Figure 2 biomolecules-10-01534-f002:**
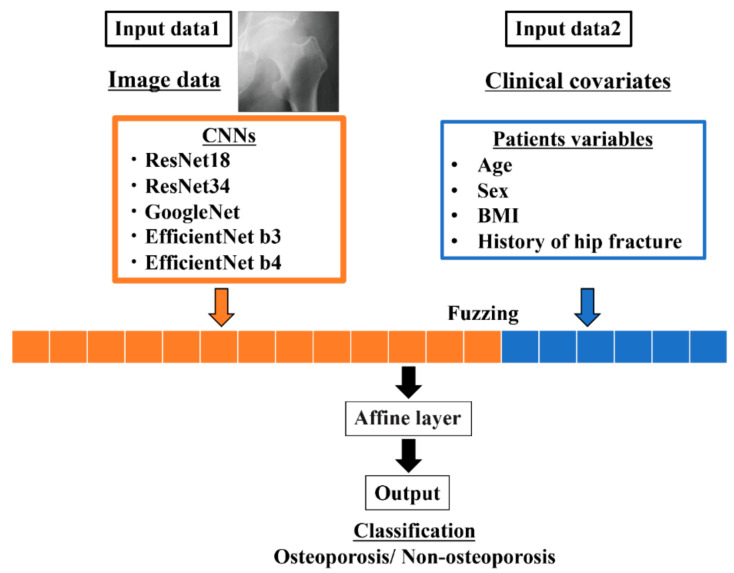
A neural-network model architecture that ensembles image data and clinical covariates.

**Figure 3 biomolecules-10-01534-f003:**
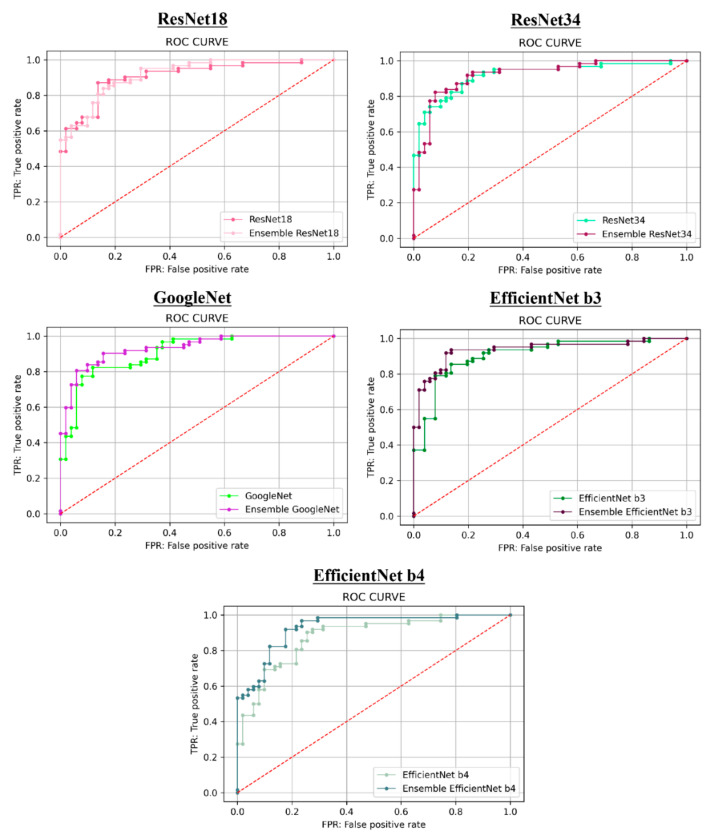
ROC curves for each of the CNN models with hip radiographs alone and the CNN analysis with hip radiographs combined with clinical covariates analysis.

**Figure 4 biomolecules-10-01534-f004:**
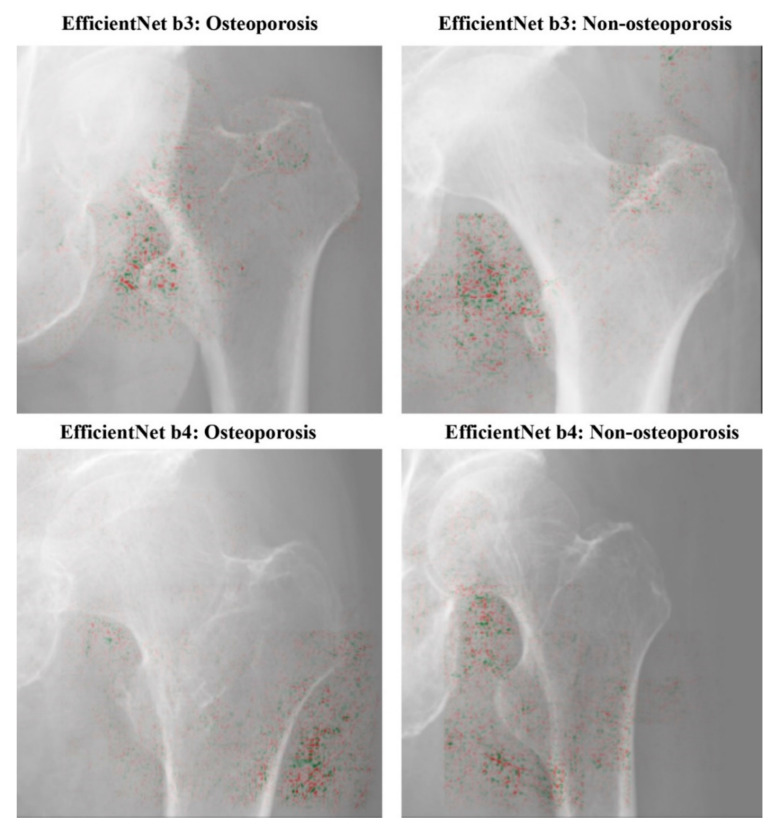
Visualization of characteristic regions of radiographs of osteoporosis a non-osteoporosis patient images on Efficientnet b3 and b4.

**Table 1 biomolecules-10-01534-t001:** Clinical and demographic characteristics of the hip radiographic dataset in this study.

	Osteoporosis	Non-Osteoporosis	*p* Value
	(T-score ≤ −2.5)	(T-score > −2.5)	
**Number of patients**	598	535	
**Sex**			<0.0001
**Female (%)**	506 (84.6)	371 (69.3)	
**Male (%)**	92 (15.4)	164 (30.7)	
**Mean age (SD, min-max)**	82.7 (8.3, 60, 100)	77.7 (9.0, 60–98)	<0.0001
**BMI (SD, min-max)**	20.1 (3.1, 13.3–29.0)	23.3 (9.0, 14.1–39.2)	<0.0001
**History of hip fracture (%)**	250 (41.8)	157 (29.3)	<0.0001

**Table 2 biomolecules-10-01534-t002:** Prediction performance on hip radiographic-image data only.

	Accuracy	Precision	Recall	Specificity	npv	F1 Score	AUC Score
**ResNet18**	0.7876	0.8654	0.7258	0.8627	0.7213	0.7895	0.9089
**ResNet34**	0.8407	0.8793	0.8226	0.8627	0.8000	0.8500	0.9203
**GoogleNet**	0.8407	0.8929	0.8065	0.8824	0.7895	0.8475	0.9064
**EfficientNet b3**	0.8407	0.8929	0.8065	0.8824	0.7895	0.8475	0.9089
**EfficientNet b4**	0.8053	0.8030	0.8548	0.7451	0.8085	0.8281	0.8786

(npv: negative predictive value).

**Table 3 biomolecules-10-01534-t003:** Prediction performance on hip radiographic images with clinical covariates.

	Accuracy	Precision	Recall	Specificity	npv	F1 Score	AUC Score
**ResNet18**	0.8407	0.8667	0.8387	0.8431	0.8113	0.8525	0.9190
**ResNet34**	0.8673	0.9273	0.8226	0.9216	0.8103	0.8718	0.9219
**GoogleNet**	0.8584	0.8966	0.8387	0.8824	0.8182	0.8667	0.9330
**EfficientNet b3**	0.8850	0.9016	0.8871	0.8824	0.8654	0.8943	0.9374
**EfficientNet b4**	0.8584	0.8594	0.8871	0.8235	0.8571	0.8730	0.9282

(npv; negative predictive value).

**Table 4 biomolecules-10-01534-t004:** Rate of change in predicted performance of ensemble model and image only by clinical variables {Ensemble model/hip radiographs alone (%)}.

	Accuracy	Precision	Recall	Specificity	npv	F1 Score	AUC Score
**ResNet18**	106.7	100.2	115.6	97.7	112.5	108.0	101.1
**ResNet34**	103.2	105.5	100.0	106.8	101.3	102.6	100.2
**GoogleNet**	102.1	100.4	104.0	100.0	103.6	102.3	102.9
**EfficientNet b3**	105.3	101.0	110.0	100.0	109.6	105.5	103.1
**EfficientNet b4**	106.6	107.0	103.8	110.5	106.0	105.4	105.6

(npv; negative predictive value).

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
