# Peer review of "Deep Learning for Osteoporosis Classification Using Hip Radiographs and Patient Clinical Covariates"

_biomolecules, 2020, doi:10.3390/biom10111534_

Round 1
Reviewer 1 Report
Using hip radiographs, this paper proposes to combine deep learning models with clinical covariates for osteoporosis classification.
As a major comment, the paper is not well written and lacks of details and precisions.
Introduction has to be improved, many references are lacking regarding AI and especially computer vison methods dealing with osteoporosis. The literature is very rich on this subject.
As among non-osteoporotic patients, there are also fractured patients, the objective of the study is not clear. Classification of osteoporosis? Classification of fractured persons? Please clarify.
The title of the paper is not appropriate. The word “Diagnosis” is used incorrectly. The paper deals only with the classification of two populations (osteoporotics and non-osteoporotics). This is not related to diagnosis of osteoporosis.
How does this study relate to the journal targeted for publication, "Biomolecules"?
Line 47:
Drawbacks cited by the authors regarding DXA are not credible. The main disadvantages concern the prediction of osteoporosis, which can be trusted up to 65%. Thus, the diagnosis of osteoporosis using DXA has to be completed by other techniques.
Line 64:
Paper [18] is not the only paper dealing with hip fractures and deep learning. Paper [18] uses machine learning and not deep learning. Paper [18] is not so recent (2013).
Line 82:
What do authors mean by segmented dataset were used. How was the segmentation achieved? Please clarify.
Section 2.4 is not clear enough.
Line 133: (Table 1)
A p-value test should be added to show if the populations are not statistically different regarding: age, BMI.
Also relative proportions of females and males should be discussed. Can this influence the results?
How did the authors take into account the imbalancement of the data to compute the different metrics (Accuracy, Precision, etc.)?
Line 135: Section 2.6
Authors should give few words about the selected CNNs and motivate their choices.
Figure 2 shows that both data (radiographs and clinical covariates) were used for the classification. However, in the text this is not clearly explained.
Moreover, it is not clear how the clinical covariates were fused to features extracted from the CNNS.
Did authors use only one CNN at a time or did they combine all features from all CNNs.
Figure 2 is not clear enough. What is behind the fully connected layers?
Please give more details.
It would be better to use the term “fuzing” rather than “concentrate”
Line 171: Section 2.8
How do authors use Grad-CAMs to show the areas that affected the network decision. This is not clear. What does mean: “where green indicated high attenuation and red indicated low attenuation”?
Line 187: Table 2
Highest values should be highlighted in bold.
Line 201: Table 3 and elsewhere
“Structured data” should be replaced by “clinical covariates”
Lin 190: Section 3.1.2
The gain due to clinical variables should be specified in percentage
Figure 3:
Captions are not appropriate.
Figure 4:
Some of the regions highlighted by the Grad-CAM are outside of the femur. Please discuss and clarify.
Line 218: Discussion
The discussion might be improved including more references in link with the subject of the paper.
Author Response
Reviewer1
Using hip radiographs, this paper proposes to combine deep learning models with clinical covariates for osteoporosis classification.
As a major comment, the paper is not well written and lacks of details and precisions.
# Reviewer1 Comment 1; - Introduction has to be improved, many references are lacking regarding AI and especially computer vison methods dealing with osteoporosis. The literature is very rich on this subject.
Author response: We thank the reviewer for the helpful comments. As you pointed out, we have added some machine learning academic papers on osteoporosis.
# Reviewer1 Comment 2; - As among non-osteoporotic patients, there are also fractured patients, the objective of the study is not clear. Classification of osteoporosis? Classification of fractured persons? Please clarify.
Author response: We thank the reviewer for the helpful comments.
As shown in the introduction, the purpose of this study was to classified the presence or absence of osteoporosis with hip radiographs using deep learning.
# Reviewer1 Comment 3; - The title of the paper is not appropriate. The word “Diagnosis” is used incorrectly. The paper deals only with the classification of two populations (osteoporotics and non-osteoporotics). This is not related to diagnosis of osteoporosis.
Author response: Thank you for your valuable opinion. We changed the title of our paper to "Deep Learning for Osteoporosis Classification Using Hip Radiographs and Patient Variables".
# Reviewer1 Comment 4; - How does this study relate to the journal targeted for publication, "Biomolecules"?
Author response: We posted in the special issue "Application of Artificial Intelligence for Medical Research" of "Biomolecules".
The purpose of this issue is "With this Special Issue, we aim to cover topics on application of artificial intelligence for medical research, in particular focusing on integrated analysis of medical data using Machine Learning and Deep Learning." The study of whether osteoporosis can be classified from hip roentgen photographs is consistent with the purpose of the special issue. Thank you for your understanding.
# Reviewer1 Comment 5; -:Drawbacks cited by the authors regarding DXA are not credible. The main disadvantages concern the prediction of osteoporosis, which can be trusted up to 65%. Thus, the diagnosis of osteoporosis using DXA has to be completed by other techniques.
Author response: Thank you for your valuable opinion. The drawbacks of DXA are the associated measurement errors and in examination costs due to the surrounding soft tissue. DXA generally has a high cost. Therefore, it is difficult to carry out DXA in poor areas. We added and revised the citations for each.
# Reviewer1 Comment 6; -Paper [18] is not the only paper dealing with hip fractures and deep learning. Paper [18] uses machine learning and not deep learning. Paper [18] is not so recent (2013).
Author response: Thank you for your valuable opinion.
As you pointed out, the reference is machine learning. The manuscript has been revised.
# Reviewer1 Comment 7; -What do authors mean by segmented dataset were used. How was the segmentation achieved? Please clarify.
Author response: Thank you for your valuable opinion. This study is to classify the presence or absence of osteoporosis from X-ray images around the hip joint. The segmented dataset mimicked the diagnostic range of osteoporosis by the DXA method. The method of segmentation in preparation for analysis is described in 2.3 Data preprocessing. We have added to the Materials and Methods section.
# Reviewer1 Comment 8; - A p-value test should be added to show if the populations are not statistically different regarding: age, BMI.
Author response: We thank the reviewer for the helpful comments.
We have added the P-values for each patient data to Table 1.
# Reviewer1 Comment 9; -Also relative proportions of females and males should be discussed. Can this influence the results?
Author response: We thank the reviewer for the helpful comments.
From table 1, it can be seen that osteoporosis has a significantly higher history of women, elderly patients, decreased BMI, and hip fractures. These results were consistent with previous studies. And these significant patient factors ensemble with the classification of osteoporosis from X-ray images will have great significance in the classification results. The above content has been added to the discussion.
# Reviewer1 Comment 10; -How did the authors take into account the imbalancement of the data to compute the different metrics (Accuracy, Precision, etc.)?
Author response: We thank the reviewer for the valuable comments.
It is as you pointed out. In this study, we selected four data as significant factors for osteoporosis. By ensemble these data with the image data, it is considered that the diagnostic accuracy is higher than the analysis of the image alone.
# Reviewer1 Comment 11; -Authors should give few words about the selected CNNs and motivate their choices.
Author response: We thank the reviewer for the valuable comments.
We have added the following sentence to 2.6: ResNet and GoogleNet, the past winning models of the ImageNet Large Scale Visual Recognition Challenge, a competition for computer-based image recognition technology, were selected as CNN. We also used EfficientNet, a state-of-the-art CNN model that is relatively smaller and faster than other CNN models.
# Reviewer1 Comment 12; -Figure 2 shows that both data (radiographs and clinical covariates) were used for the classification. However, in the text this is not clearly explained.
Moreover, it is not clear how the clinical covariates were fused to features extracted from the CNNS.
Author response: We thank the reviewer for the valuable comments. We have added a new section "2.7. Architecture of the ensemble model" and added a description of the ensemble model structure.
# Reviewer1 Comment 13; - Did authors use only one CNN at a time or did they combine all features from all CNNs.
Author response: Yes. We used only one CNN at a time.
# Reviewer1 Comment 14; - Figure 2 is not clear enough. What is behind the fully connected layers?
Please give more details.
It would be better to use the term “fuzing” rather than “concentrate”
Author response: We thank the reviewer for the valuable comments.
We corrected the phrase and replaced Figure 2 with a new figure.
# Reviewer1 Comment 15; -How do authors use Grad-CAMs to show the areas that affected the network decision. This is not clear. What does mean: “where green indicated high attenuation and red indicated low attenuation”?
Author response: We thank the reviewer for the valuable comments.
The description of areas of interest in deep learning has been revised as follows: All visualizations used iridescent map projection. Within the region of interest, green showed high attenuation and red showed low attenuation.
# Reviewer1 Comment 16; -Highest values should be highlighted in bold.
Author response: We thank the reviewer for the valuable comments.
In tables 2 and 3, the highest value has been changed to highlight in bold.
# Reviewer1 Comment 17; - Table 3 and elsewhere
“Structured data” should be replaced by “clinical covariates”
Author response: We thank the reviewer for the valuable comments.
We replaced “Structured data” with “clinical covariates”.
# Reviewer1 Comment 18; - Section 3.1.2 The gain due to clinical variables should be specified in percentage
Author response: We thank the reviewer for the valuable comments.
We have added Table 4 to show the rate of change by the ensemble model.
# Reviewer1 Comment 19; -Figure 3 Captions are not appropriate.
Author response: We thank the reviewer for the valuable comments.
We have modified Figure 3.
# Reviewer1 Comment 20; -Figure 4:
Some of the regions highlighted by the Grad-CAM are outside of the femur. Please discuss and clarify.
Author response: We thank the reviewer for the valuable comments.
It is as you pointed out. The following text has been added to the discussion; The area identified by Grad-CAM coincided with the area around the lesser trochanter. This may provide a reason for the diagnosis of osteoporosis. On the other hand, some of the areas highlighted by Grad-CAM were on the outside of the femur. The output of Grad-CAM is purely for reader's references but doesn't directly imply the prediction itself[17]. Because our model is trained for classification but not for localization, thus it might focus some inappropriate regions that Grad-CAM shows, nonetheless the prediction is correct or not. In the future, it will be necessary to further increase the number of cases and examine different images such as image size and soft tissue contrast.
# Reviewer1 Comment 21; -Line 218 Discussion
The discussion might be improved including more references in link with the subject of the paper.
Author response: We thank the reviewer for the valuable comments.
We have upgraded this paper by adding many citations.

Reviewer 2 Report
1. The authors investigated that the lesser trochanter is the major area of focused visualization by EfficientNet b3. The distinct deep-learned areas may impact the efficiency of deep-learning. Dose different deep learning identify different areas of hip joint to diagnose osteoporosis?
2. Convolutional Neural Network (CNN) is a model to identify image. Can the author explain the work of CNN to diagnose osteoporosis with image and clinical data? Why is the efficiency of deep learning with image and structured data better?
3. What role of Structured data do play in this study to improve the efficiency of deep learning?
Author Response
The manuscript has been carefully rechecked and appropriate changes have been made in accordance with the reviewers’ suggestions. The major changes made are highlighted in red font in the revised document. The responses to their comments have also been prepared and are attached herewith.
Reviewer2
- The authors investigated that the lesser trochanter is the major area of focused visualization by EfficientNet b3. The distinct deep-learned areas may impact the efficiency of deep-learning. Dose different deep learning identify different areas of hip joint to diagnose osteoporosis?
Author response: We thank the reviewer for the valuable comments.
Added the image of Gided gradcam of Efficientnet b4 in addition to Efficientnet b3. It referred to other different bone regions of the femur. Visualization will need to be done on other CNNs in the future.
# Reviewer2 Comment 2; - Convolutional Neural Network (CNN) is a model to identify image. Can the author explain the work of CNN to diagnose osteoporosis with image and clinical data? Why is the efficiency of deep learning with image and structured data better?
# Reviewer2 Comment 3; - What role of Structured data do play in this study to improve the efficiency of deep learning?
Author response: We thank the reviewer for the helpful comments.
Our research models are classifier that combines image features and patient factors with a neural network. It can be inferred that the accuracy has improved because it is possible to make inferences while simultaneously considering important information related to identification (patient clinical factors) that cannot be extracted from images alone.
The above content has been added to the discussion.

Reviewer 3 Report
The authors propose the use of Deep Learning for Osteoporosis Diagnosis Using Hip Radiographs and Patient Variables. The topic of the paper is of much interest, the conducted research seems of quality, and the paper is in general well written.
However, before recommending it for publication, the following issues should be addressed.
- For validation, the authors divided the dataset randomly into training, validation, and testing sets in a ratio of 8:1:1. This technique is known as Holdout method. The authors should use a stronger validation procedure such as k-fold cross-validation. And explicitly state what cross validation procedure they applied, how, and most importantly, why. Also they need to provide additional detail about the splitting process, and
- Authors claim they used data augmentation, by randomly rotated and flipping the images, but do not explain appropriately when they apply this process or how they use this strategy, if the dataset was increased in size with this strategy... This must be clarified.
- The authors should show the Learning Curves for training and validation, so the reader is able to evaluate properly the performance of the models.
- The authors only only a paper with previously published results of Hip diagnosys using radiographs: (Badgeley, M.A.; Zech, J.R.; Oakden-Rayner, L.; Glicksberg, B.S.; Liu, M.; Gale, W.; McConnell, M.V.; 356 Percha, B.; Snyder, T.M.; Dudley, J.T. Deep learning predicts hip fracture using confounding patient and 357 healthcare variables. npj Digit. Med. 2019, 2, 31). In this paper the authors claim they did not optimize model hyper-parameters or model selection. So it is very hard to evaluate the quality of the obtained results with reference of previously published work. The authors should give a clear perspective of the results obtained in previous works in similar scenarios, summarizing the previously obtained numerical results. Or if this is the case, explain more clearly that there are no previous results that can be compared to this work and support this statement with evidence.
- I think table 1 does not provide enough info of the population, the authors may add in the paper or as supplementary material additional information with the distribution of age, BMI and history of hip fracture per gender, and a correlation analysis of age, BMI and history of hip fracture with the target variables.
- Finally, I would recommend a general language and flow improvement in the manuscript, in order to make it more clear and easy to follow.
Author Response
The manuscript has been carefully rechecked and appropriate changes have been made in accordance with the reviewers’ suggestions. The major changes made are highlighted in red font in the revised document. The responses to their comments have also been prepared and are attached herewith.
Reviewer3
The authors propose the use of Deep Learning for Osteoporosis Diagnosis Using Hip Radiographs and Patient Variables. The topic of the paper is of much interest, the conducted research seems of quality, and the paper is in general well written.
However, before recommending it for publication, the following issues should be addressed.
# Reviewer3 Comment 2; - For validation, the authors divided the dataset randomly into training, validation, and testing sets in a ratio of 8:1:1. This technique is known as Holdout method. The authors should use a stronger validation procedure such as k-fold cross-validation. And explicitly state what cross validation procedure they applied, how, and most importantly, why. Also they need to provide additional detail about the splitting process
Author response: We thank the reviewer for the precious comments.
It is as you pointed out. Cross-validation could not be used in this paper due to time and cost constraints. In the future, it will be necessary to prepare the environment and study generalization by cross-validation.
# Reviewer3 Comment 2; - Authors claim they used data augmentation, by randomly rotated and flipping the images, but do not explain appropriately when they apply this process or how they use this strategy, if the dataset was increased in size with this strategy... This must be clarified.
Author response: We thank the reviewer for the helpful comments. Data Augmentation was applied only to training image data. In the learning process, data augmentation was processed when the images were taken out as a batch. Each was processed with a 50% chance. The above contents have been added to "2.6. CNN model architecture".
# Reviewer3 Comment 3; -The authors should show the Learning Curves for training and validation, so the reader is able to evaluate properly the performance of the models.
Author response: We thank the reviewer for the helpful comments. All models analyzed a maximum epoch of 100 as proof that the training model reached libido. In addition, we chose the early stop method, which stops learning if the validation error is not updated 20 times in a row.  The above was added to "2.6. CNN model architecture".
# Reviewer3 Comment 4; -The authors only only a paper with previously published results of Hip diagnosys using radiographs: (Badgeley, M.A.; Zech, J.R.; Oakden-Rayner, L.; Glicksberg, B.S.; Liu, M.; Gale, W.; McConnell, M.V.; 356 Percha, B.; Snyder, T.M.; Dudley, J.T. Deep learning predicts hip fracture using confounding patient and 357 healthcare variables. npj Digit. Med. 2019, 2, 31). In this paper the authors claim they did not optimize model hyper-parameters or model selection. So it is very hard to evaluate the quality of the obtained results with reference of previously published work. The authors should give a clear perspective of the results obtained in previous works in similar scenarios, summarizing the previously obtained numerical results. Or if this is the case, explain more clearly that there are no previous results that can be compared to this work and support this statement with evidence.
Author response: We thank the reviewer for the precious comments. In our paper, we did not adjust the hyperparameters with reference to the paper you pointed out.
Since the adjustment of hyperparameters is expensive and takes a long time for one learning, the process of adjusting the parameters many times and re-learning cannot be realized in terms of time and cost. In the future, it will be necessary to prepare the environment and consider it.
# Reviewer3 Comment 5; -I think table 1 does not provide enough info of the population, the authors may add in the paper or as supplementary material additional information with the distribution of age, BMI and history of hip fracture per gender, and a correlation analysis of age, BMI and history of hip fracture with the target variables.
Author response: We thank the reviewer for the helpful comments. 
It is as you pointed out. Table 1 was modified to add age distribution and BMI distribution.
# Reviewer3 Comment 6; -Finally, I would recommend a general language and flow improvement in the manuscript, in order to make it more clear and easy to follow.
Author response: We thank the reviewer for the helpful comments.
The entire manuscript was reviewed and revised again.

Round 2
Reviewer 1 Report
This is a new resubmission of paper 939141-v1.
Although most of my further comments were taken into account, the paper still needs to be improved before publication.
The paper has to be English proofread.
In Table 1, it is not clear whether the p-values are given for women or for man or for both mixed populations.
In Table 3, how can the values be superior to 100%?
Regarding my previous comment, the answer is not clear and not convincing.
# Reviewer1 Comment 10; -How did the authors take into account the imbalancement of the data to compute the different metrics (Accuracy, Precision, etc.)?
According to the data, there are 598 osteoporotic and 535 non-osteoporotic. The data are imbalanced and the metrics used to compute accuracy, precision, etc. are no more valid. Other formula should be used.
Author Response
Although most of my further comments were taken into account, the paper still needs to be improved before publication.
# Reviewer1 Comment 1; - The paper has to be English proofread.
Author response: We thank the reviewer for the valuable comments.
We have been checked for English proofreading. We will attach the certificate.
# Reviewer1 Comment 2; - In Table 1, it is not clear whether the p-values are given for women or for man or for both mixed populations.
Author response: We thank the reviewer for the valuable comments. The p-value was given for both mixed populations. We have modified Table 1.
# Reviewer1 Comment 3; - In Table 3, how can the values be superior to 100%?
Author response: We thank the reviewer for the valuable comments.
Tables 3 and 4 had been replaced. We have modified Table3 and 4.
# Reviewer1 Comment 4; -Regarding my previous comment, the answer is not clear and not convincing.
How did the authors take into account the imbalancement of the data to compute the different metrics (Accuracy, Precision, etc.)?
According to the data, there are 598 osteoporotic and 535 non-osteoporotic. The data are imbalanced and the metrics used to compute accuracy, precision, etc. are no more valid. Other formula should be used.
Author response: We thank the reviewer for the valuable comments.
This study was performed on 598 osteoporotic and 535 non-osteoporotic data. This data is practically well balanced. Based on that, the test data was also calculated by adjusting the ratio to 598 osteoporotic and 535 non-osteoporotic.

Reviewer 2 Report
Accept in present form
Author Response
We thank the reviewer for the valuable comments.
Thank you for your great effort in peer review.
Reviewer 3 Report
The authors have improved the paper a lot, and have addressed most of my concerns.
However, I think some further modifications are still required:
The authors claim no previous work has used hip radiographs for osteoporosis diagnosis based with DL, and only one old paper has proposed a system based on SVM.
The authors claim they have comparable results to this work, and state that the experiment they performed is more robust. They need to expand on this, in a specific way, showing the numerical results of both experiments, the specific number of cases and so on.
Also, I have found some relevant work the authors may want to reference:
- Lee, Ju Hwan, et al. "Diagnosis of osteoporosis by quantification of trabecular microarchitectures from hip radiographs using artificial neural networks." Bio-Inspired Computing-Theories and Applications. Springer, Berlin, Heidelberg, 2014. 247-250.ç
- Wani, Insha Majeed, and Sakshi Arora. "Deep Neural Networks for Diagnosis of Osteoporosis: A Review." Proceedings of ICRIC 2019. Springer, Cham, 2020. 65-78.
- Rastegar, Sajjad, et al. "Radiomics for classification of bone mineral loss: A machine learning study." Diagnostic and interventional imaging 101.9 (2020): 599-610.
- Sah, Alexander P., et al. "Correlation of plain radiographic indices of the hip with quantitative bone mineral density." Osteoporosis international 18.8 (2007): 1119-1126.
Author Response
The authors have improved the paper a lot, and have addressed most of my concerns.
However, I think some further modifications are still required:
# Reviewer3 Comment 1; -The authors claim no previous work has used hip radiographs for osteoporosis diagnosis based with DL, and only one old paper has proposed a system based on SVM. The authors claim they have comparable results to this work, and state that the experiment they performed is more robust. They need to expand on this, in a specific way, showing the numerical results of both experiments, the specific number of cases and so on.
Author response: We thank the reviewer for the helpful comments.
The results reported in the literature have been added for comparison with our study.
# Reviewer3 Comment 2; -
Also, I have found some relevant work the authors may want to reference:
Lee, Ju Hwan, et al. "Diagnosis of osteoporosis by quantification of trabecular microarchitectures from hip radiographs using artificial neural networks." Bio-Inspired Computing-Theories and Applications. Springer, Berlin, Heidelberg, 2014. 247-250.ç
Wani, Insha Majeed, and Sakshi Arora. "Deep Neural Networks for Diagnosis of Osteoporosis: A Review." Proceedings of ICRIC 2019. Springer, Cham, 2020. 65-78.
Rastegar, Sajjad, et al. "Radiomics for classification of bone mineral loss: A machine learning study." Diagnostic and interventional imaging 101.9 (2020): 599-610.
Sah, Alexander P., et al. "Correlation of plain radiographic indices of the hip with quantitative bone mineral density." Osteoporosis international 18.8 (2007): 1119-1126.
Author response: We thank the reviewer for the helpful comments. With reference to the proposed paper, we added it to the cited references and added it to the discussion section.